# Animal Models for Human Polycystic Ovary Syndrome (PCOS) Focused on the Use of Indirect Hormonal Perturbations: A Review of the Literature

**DOI:** 10.3390/ijms20112720

**Published:** 2019-06-03

**Authors:** Youngjae Ryu, Sung Woo Kim, Yoon Young Kim, Seung-Yup Ku

**Affiliations:** 1Biomedical Research Institute, Seoul National University Hospital, Seoul 03080, Korea; dragonkai@naver.com (Y.R.); yoonykim@snu.ac.kr (Y.Y.K.); 2Department of Obstetrics and Gynecology, College of Medicine, Seoul National University, Seoul 03080, Korea; byulbi81@snu.ac.kr

**Keywords:** polycystic ovary syndrome, animal models, ovary, pathogenesis

## Abstract

Hormonal disturbances, such as hyperandrogenism, are considered important for developing polycystic ovary syndrome (PCOS) in humans. Accordingly, directly hormone-regulated animal models are widely used for studying PCOS, as they replicate several key PCOS features. However, the pathogenesis and treatment of PCOS are still unclear. In this review, we aimed to investigate animal PCOS models and PCOS-like phenotypes in animal experiments without direct hormonal interventions and determine the underlying mechanisms for a better understanding of PCOS. We summarized animal PCOS models that used indirect hormonal interventions and suggested or discussed pathogenesis of PCOS-like features in animals and PCOS-like phenotypes generated in other animals. We presented integrated physiological insights and shared cellular pathways underlying the pathogenesis of PCOS in reviewed animal models. Our review indicates that the hormonal and metabolic changes could be due to molecular dysregulations, such as upregulated PI3K-Akt and extracellular signal-regulated kinase (ERK) signalling, that potentially cause PCOS-like phenotypes in the animal models. This review will be helpful for considering alternative animal PCOS models to determine the cellular/molecular mechanisms underlying PCOS symptoms. The efforts to determine the specific cellular mechanisms of PCOS will contribute to novel treatments and control methods for this complex syndrome.

## 1. Introduction

Various animal models have been developed and studied for the human polycystic ovary syndrome (PCOS) for more than 60 years. However, the etiology of PCOS is still unclear because of its complex manifestation as a syndrome and limitations of translational studies using animals.

Although a huge gap exists between humans and laboratory animals, which are mainly rodents, with respect to their reproductive physiology (e.g., differences in ovulation number and patterns, released hormone profile, sensitivity to hormones, behavioural styles, and anatomy of the organs), studies using animal models are essential to explore the pathophysiology of PCOS in vivo.

According to Rotterdam’s criteria for PCOS diagnosis [1], the key features of PCOS that should be manifested in the animal models are hyperandrogenism, too many antral follicles and abnormal estrous cycle (amenorrhea or oligomenorrhea in case of non-human primates). The most suitable animal PCOS model would be naturally occurring PCOS-like animals. However, laboratory rodents, unlike humans, do not develop PCOS-like phenotypes naturally, although the persistence of ovarian follicles with a prolonged estrous period has been reported in natural housed rats [2]. To date, natural PCOS-like features have not been reported in rodents. Natural development of polycystic ovary (PCO) and related hormone disturbances with clinical symptoms, has been observed in animals such as cow, pig, and dog; however, these animals did not generally show hyperandrogenism. Naturally occurring hyperandrogenism has been reported in non-human primates. Abbott et al. (2017) reported PCOS-like traits in naturally hyperandrogenic female rhesus monkeys (*Macaca mulatta*) [3]. However, the use and screening of non-human primates for PCOS studies have many limitations.

Therefore, to study PCOS with laboratory animals, various artificial methods have been used; prenatally or postnatally hormone-treated (i.e., excess androgen or estrogen) animals are extensively used. These animal models can manifest the key features of PCOS, such as hyperandrogenism, those with excess antral follicles, prolonged or irregular estrous cycle, and metabolic symptoms. Moreover, prenatal perturbation using androgens has demonstrated transgenerational development of PCOS phenotypes in the offspring. However, to define the specific pathogenesis of PCOS and explore control/treatment methods, it is undoubtedly valuable to establish other methods along with the hormonal intervention models.

Thus, herein we focused on highlighting up-to-date animal PCOS models that use indirect hormonal interventions and summarizing suggested/discussed mechanisms of the pathogenesis of the developed PCOS-like phenotypes in those models. Furthermore, we have briefly summarized naturally developed PCO and relevant mechanisms in other animals to search for shared pathophysiology with respect to polycystic ovarian morphology, hormonal alteration, and accompanying metabolic syndromes. We have also discussed shared cellular/molecular mechanisms underlying the pathogenesis of PCOS in animals on the basis of a literature review and suggested further directions for the use of animal PCOS models.

## 2. Methods

We searched the literature up to June 2018 by using PubMed (https://www.ncbi.nlm.nih.gov/pubmed/) and Google Scholar (https://scholar.google.com/). The papers were searched using the following terms: ‘PCOS’, ‘polycystic ovary syndrome’, ‘animal model’, ‘mouse’, ‘rat’, ‘primate’, ‘transgenic’, ‘chemical’, ‘ovarian dysfunction’, ‘testosterone’, ‘androgen’, and ‘pathogenesis’. We excluded studies with androgen-, estrogen-, or other directly hormone-treated animal PCOS models in which either prenatal or postnatal administration was performed. In addition, we excluded studies in which evaluation of the ovarian morphology and serum androgen levels was not performed. However, we included the papers when following or similar studies using same experimental methods backed those evaluations up.

Furthermore, to search for new aspects of pathogenesis of PCOS in other species, we used keywords such as ‘cystic ovarian disease’, ‘COD’, ‘polycystic ovary’, ‘animal’, and ‘veterinary’. Among the papers, we selected representative literature to discuss the relevant mechanisms.

Human PCOS patients with more than 12 cystic follicles per ovary or ovarian volume over 10 mL, diagnosed using ultrasonography, are confirmed to have ‘excess dominant follicles’ [4]. In this review, we use the term ‘PCO’ or for animals on the basis of histological section images and sentences in the reviewed literature that describe those with excess numbers of antral follicles. In addition, we used ‘hyperandrogenism’ as the status presented by significantly elevated serum testosterone concentration when compared with the control group. All the animals described in this review are female, except when especially noted.

### 2.1. Animal Polycystic Ovary Syndrome (PCOS) Models Induced by Direct Hormonal Interventions

Because one of a clinical hallmark of PCOS patients is hyperandrogenism, to induce PCOS-like phenotypes in laboratory animals, direct androgen treatment methods have been widely used with agents such as testosterone propionate (TP), dihydrotestosterone (DHT), or dehydroepiandrosterone. Other hormonal agents such as estradiol valerate (EV), glucocorticoid, human chorionic gonadotropin (hCG), and anti-Müllerian hormone (AMH) have also been used. These hormones induce PCOS-like phenotypes by not only postnatal administration but also prenatal administration in rodents, sheep, and non-human primates.

Many well-written reviews have already summarized these types of animal PCOS models [5,6,7,8,9,10,11,12,13,14]. In this review, we aimed to introduce mainly indirect hormonal methods for developing PCOS-like phenotypes in animals; therefore, we have provided a short description of hormone-induced animal PCOS models.

On the basis of the results obtained using androgen receptor- Knockout (KO) mice, excess androgen is sufficient to induce an irregular estrous cycle, excess dominant follicles and other metabolic symptoms in mice. Although further studies are needed to check whether the effect of androgen is mainly on the neuronal regulation or ovaries, it is clear that exposure to high androgen levels is sufficient to induce PCOS-like manifestations in rodents.

Animals such as sheep and monkeys exposed to androgens prenatally have also developed hallmarks of PCOS phenotypes. Luteinizing hormone (LH) pulse frequency is also elevated in prenatally androgen exposed sheep and monkeys [15]. In other animals, such as sheep, and rhesus monkey, prenatal androgen exposure induced hyperandrogenism and oligo-anovulation in the offspring [9]. Androgen exposure during the gestation stage re-programs the hypothalamic-pituitary-gonadal (HPG) axis through epigenetic changes or neuronal rearrangements [16,17]. Recently, mice treated with Anti-Müllerian hormone (AMH) prenatally showed acyclic estrous and elevated serum testosterone and LH levels [18]. The authors indicated that excess AMH exposure during the late period of pregnancy (embryonic day 16.5 to 18.5) leads to hyperactivation of the gonadotropin-releasing hormone (GnRH) neurons in the offspring, resulting in fluctuations in LH release from the pituitary gland. As a result, this model induced a masculinization of the exposed female fetus and a PCOS-like reproductive and neuroendocrine phenotype in adulthood.

#### Limitations and Difficulties of Hormone-Treated Animal Models

Androgen-treated models yield good reproduction results and manifestation of the relevant PCOS symptoms in animals. However, direct hormonal interventions may not be suitable for finding a basal and common pathway for its pathogenesis. However, the experiments demonstrated that androgen exposure can alter endocrine homeostasis and lead to the development of ovarian dysfunction as well as its accompanying metabolic symptoms. There are many mechanisms driving increased androgen production in PCOS include increased LH stimulation resulting from abnormal LH secretory dynamics and increased LH bioactivity, and hyperinsulinemia due to insulin resistance. Considering that PCOS is a hyperandrogenic disorder [19,20], hyperandrogenism is one of the most important features to be elaborately manifested in the animal model. In addition, considering both human males and females may not be exposed to such excessive hormonal disturbance in their lives.

LH stimulation and alteration of GnRH neuron activation may be the fundamental reasons for hyperandrogenism in PCOS patients [21]. However, increased insulin levels (hyperinsulinemia) and insulin resistance have been suggested as the key triggers for the development of PCOS [22,23]. Other factors such as alterations in AMH and Follicle-stimulating hormone (FSH) levels may induce the occurrence of PCOS [24]. Indeed, all these factors are related to PCOS pathogenesis on the basis of both clinical data and animal experiments [25,26]. However, the shared molecular pathway/mechanism and causal relationship regarding the pathogenesis of PCOS are still unclear. Therefore, animal PCOS models that do not use direct hormonal intervention may be useful.

### 2.2. Animal Models with PCOS or Similar Symptoms Induced by Indirect Hormonal Perturbations

#### 2.2.1. Genetically Engineered or Genetic Animal Models Including PCOS-Like Symptoms

To understand the role of specific genes and their pathways related to the development of PCOS, various genetically modified rodents have been studied. We reviewed literature focused on elevated serum testosterone levels and the polycystic ovarian appearance in the animals. In addition, we noted suggested/discussed mechanisms responsible for the developed phenotypes (Table 1).

##### PCOS-Like Phenotypes Caused by Metabolic Dysfunctions

In the numerous genetically manipulated or spontaneously mutated rodent models, the rodent strains or transgenic mice that show dysregulations in lipid or glucose metabolism commonly manifest ovarian cysts and metabolic dysfunctions. The New Zealand obese mice and Zucker (fa/fa) rat, which are both spontaneous genetic obesity rodent models, commonly present hyperinsulinemia, ovarian cysts, and poor fertility. However, these strains did not show elevated (or even significantly decreased) serum testosterone or LH levels [33,34,35]. Because leptin receptor mutations and reduced leptin signalling (leptin resistance) were observed in these rodents [36,37], it is assumed that leptin-mediated LH release was compromised and, therefore, synthesis of androgen by LH stimulation was not accelerated. These findings suggest that hyperinsulinemia and cystic ovarian morphology induced by metabolic abnormalities do not cause hyperandrogenemia, at least, in rodents. Nevertheless, species difference should be taken into account. It is well established that increased circulating insulin levels cause or contribute to hyperandrogenism in women with PCOS, as opposite to rodents, in at least two important ways, by stimulating increased ovarian androgen production, and by inhibiting hepatic Sex hormone-binding globulin (SHBG) production [38]. Honnma et al. indicated that downregulated ovarian adiponectin levels may related to ovarian dysfunction, although serum adiponectin level was significantly higher in the Zucker (fa/fa) rats than in the control group. However, a recent study showed that overexpression of adiponectin in mice did not ameliorate PCO morphology as well as the estrous cycle induced by DHT administration [39]. Kajihara et al. reported increased Forkhead box O (FOXO)-1 expression in the ovary tissue of Zucker (fa/fa) rats, especially in the granulosa cells. FOXOs regulate cell proliferation, response to oxidative stress, and downstream targets of the phosphatidylinositol 3-kinase (PI3K)–Akt pathway [40].

On the other hand, JCR:LA-cp rats that also had a defect in the leptin receptor (LepR) and concurrent hyperleptinemia showed a spontaneous increase in serum testosterone levels, irregular estrous cycle, cystic follicles, hyperinsulinemia, obesity, and dyslipidemia [41]. Although increased insulin level is considered a major regulatory factor of ovarian androgen genesis [42], the discrepancies among the obese rodent models (New Zealand obese mouse, Zucker rat, and JCR:LA-cp rat) may not elicit simple connections between insulin level and hyperandrogenism. Significantly increased leptin levels were detected in both the New Zealand obese mice and Zucker (fa/fa) rats [36,43]. Therefore, it is possible that the frequency of LH stimulation or expression of ovarian enzymes related to steroidogenesis may differ between those models. A previous study has reported that the increased uptake of non-esterified fatty acid in the ovary of JCR:LA-cp rats is correlated to the serum testosterone level [44]. These results suggest that intracellular ovarian lipotoxicity may be related to androgen production. The degree of fatty acid-mediated oxidation stress to the ovary may differ among the obese rodent models.

Mice lacking the insulin receptor (IR) and LepR in hypothalamic pro-opiomelanocortin neurons (POMC; IR/LepR^POMC^) manifested prolonged estrous cycle, hyperinsulinemia, hyperleptinemia, increased body fat mass, and elevated serum LH and testosterone levels [45,46]. However, the mice did not develop typical PCO morphology. In addition, adipose dysfunction, macrophage infiltration, and elevated cytokine levels (including IL-1β and IL-6) in adipose tissues such as perigonadal fat were observed in this model [29]. On the basis of these results, elevated LH stimulation and hyperinsulinemia may be related to hyperandrogenism; however, the increased insulin and serum testosterone levels did not result in the development of prominent follicular cysts in the IR/LepR^POMC^ mice.

Another transgenic obese mouse, Mito-ob, that overexpressed prohibitin in adipocytes showed the formation of ovarian cysts without any significant changes in the serum levels of insulin, leptin, LH, and testosterone [47]. Instead, increased body fat mass and serum estradiol level were observed in the Mito-ob mice. Thus, increased estradiol levels or overgrowth of the periovarian adipose tissue may be the reasons for the cystic ovarian morphology in this model.

Mice overexpressing the human insulin-like growth factor-I (IGF-I) controlled by the LH receptor (LHr) expression (LHr-hIGF-I) showed ovarian cysts and elevated serum testosterone levels. The LH level was decreased, and the estradiol level was increased, but not significantly, in this model. Upregulated IGF-I production resulted in elevated serum testosterone levels in all LHr-hIGF-I female mice; however, only 40% manifested PCO morphology, even though the histomorphological characteristics of the cysts differed from those of human PCOS patients. Therefore, the authors suggested that increased IGF-I expression may be a direct cause of hyperandrogenism in this model.

Protein kinase B-β (PKBβ/Akt2) KO mice manifested severe ovarian cysts (not the typical PCO morphology), hyperinsulinemia, and hyperandrogenemia at 90 weeks of age [48]. The serum LH level did not differ significantly when compared with the wild-type group. In addition, marked extracellular signal-regulated kinase (ERK) as well as cAMP response element-binding protein expressions and fat (lipid) accumulation aside to the ovary were observed. The 20-week-old PKBβ KO mice did not show increased serum insulin or testosterone levels. Restuccia et al. (2012) indicated that upregulated ERK signalling due to the loss of PKBβ and lipid accumulation may induce hyperandrogenism and ovarian cysts. However, Akt2 KO mice showed hyperinsulinemia and insulin intolerance at 8 weeks of age [49]; therefore, further studies are needed to clarify the mechanisms underlying the PCOS-like appearance by using this model.

Lan et al. (2017) reported that Pten^fl/fl^–Cyp17iCre (tPtenMT) mice in which *Pten* was selectively deleted in theca cells presented a prolonged estrous cycle, formation of numerous antral follicles, and elevated serum testosterone levels [50]. The theca cells of tPtenMT showed increased Akt phosphorylation and upregulated expressions of *Lhcgr*, *Star*, *Cyp11a1*, and *Cyp17a1*. In addition to the upregulation of *Lhcgr* expression, which means a more elevated response to LH, the authors reported that the phosphorylation of FOXO-1 was increased in the ovaries of the tPtenMT mice and augmented by LH administration. Moreover, *Lhcgr* may be a direct downstream target gene of FOXO-1 in the theca cells. This is an important finding: Dysregulation of the PI3K-Akt pathway and related signalling, consistently implied by the other aforementioned genetic models, can change the responsiveness to LH, resulting in hyperandrogenism.

##### PCOS-Like Phenotypes Induced by Changes in the Endocrine System

Specific deletion or overexpression of endocrine system-related genes results in the occurrence of PCOS-like features in animals. Multiple cystic follicles with haemorrhagic cysts and elevated LH levels were reported in estrogen receptor-α (ERα) KO mice by Couse et al. [51]. The ERα KO mice showed significantly increased testosterone and estradiol levels. However, ERβ KO mice did not show the same elevated levels [52]. Expression of *Lhb* in the pituitary gland and *Cyp17* and *Cyp19* in the ovary was upregulated in the ERα KO mice. Administration of the GnRH antagonist suppressed the symptoms in these mice. In addition, Lee et al. (2009) demonstrated that theca cell-specific ERα KO (thEsr1 KO), mice using 17β-hydroxylase a1 (*Cyp17*-iCre), developed PCOS-like phenotypes such as irregular estrous cycle, cystic ovaries (induced by pregnant mare’s serum gonadotropin and hCG injection), and increased serum testosterone levels. However, the LH level was significantly decreased, or even undetectable, in the mice [32]. Moreover, ERβ KO mice with the bLHβ–CTP transgene (estrogen receptor-β KO with increased LH level; bLHβ–CTP mice will be described below) presented suppressed development of cystic follicles and reduced testosterone levels when compared with wild-type bLHβ–CTP transgenic mice [53]. It is noteworthy that a loss of ERα function leads to dysregulation of LH release and an increase in *CYP17* expression in the ovary, resulting in hyperandrogenism. It can be postulated that the development of PCOS-like phenotypes is related to stimulation via ERβ, which is observed mainly in granulosa cells, and impaired functions of ERα, which is expressed in the theca or interstitial cells of the ovary as well as the hypothalamus.

Aromatase KO (Ar KO) mice developed haemorrhagic ovarian cysts, increased serum LH levels, and extensively elevated testosterone levels [54]. This model is consistent with the letrozole-treated model (described later). Interestingly, haemorrhagic ovarian cysts were not prominent in the letrozole-treated rodent model, unlike in ERα KO as well as Ar KO mice [55]. The Ar KO mice showed increased body weight, fat accumulation, and impaired lipid metabolism, such as hypercholesterolemia [56]. Estrogen replacement inversed the described symptoms in the Ar KO mice. The increase in LH release caused by estrogen depletion stimulates androgen synthesis in the theca cells, and elevated androgen levels mediate metabolic symptoms in the animals. On the basis of the results obtained using the ER KO and Ar KO mice, the formation of hyperemic ovarian cysts (haemorrhagic cysts observed in these models) may be due to the absence of estrogenic functions and dysregulation of LH stimulation in the ovary.

Risma et al. developed transgenic mice that chronically expressed the LH hormone by using the chimeric bovine LHβ gene ligated into the coding region of the carboxyl terminal peptide (CTP) of the hCG β subunit [30]. The bLHβ-CTP mice presented significantly elevated serum testosterone and estradiol levels. In addition, PCO morphology with haemorrhagic or fluid-filled cysts and anovulation was observed. The elevated serum testosterone and LH levels were observed before puberty in this model [31]. Kero et al. (2003) reported that bLHβ-CTP mice showed increased body weight with fat mass accumulation, hyperinsulinemia, hyperleptinemia, and significantly elevated serum corticosterone levels at 3 to 5 months of age [57]. In addition, they observed reduced thermogenesis in the brown adipose tissue in this model. Ovariectomized bLHβ-CTP mice did not show a significant gain in body weight when compared with the control group. Although the data time points could not be matched in the papers, these results demonstrate that increased steroidogenesis induced by chronically elevated LH levels results in the development of PCO morphology at an early stage (from 3 weeks of age) with accompanying metabolic symptoms.

Transgenic mice that overexpressed inhibin α-subunit also developed severe, but not typical PCO, ovarian cysts, and elevated serum testosterone levels [58]. The serum estradiol level was decreased, and the rate of cyst formation was gradually increased with senescence. Because inhibin stimulates LH-induced androgen synthesis in ovarian theca cells [59], hyperandrogenemia in this model is reasonable. However, based on the relatively long observation period (more than 12 months), the characteristics of the ovarian cysts (few and very large follicular cysts in the ovary) may not match other typical androgen-induced PCO morphologies reported in other studies.

##### Other Transgenic Rodent Animal Models

The plasminogen activator inhibitor-1 (PAI-1) expressing mouse, another transgenic animal model with PCO morphology and hyperandrogenemia, was reported [60]. Although the authors did not examine other factors related to PCOS, a hypertrophied theca cell layer of the ovary and highly expressed PAI-1 in the granulosa cells were described. Furthermore, considering that the PAI-1 overexpressed mouse showed hyperinsulinemia and PAI-1 is closely related to metabolic syndromes, the effects of metabolic alteration may not exclude elevated testosterone levels as well as PCO morphology [61,62].

Mice overexpressing the nerve growth factor (NGF) selectively with the promoter 17α-hydroxylase/C_17–20_ lyase (17NF mice) manifested irregular estrous cycle, hyperinsulinemia, increased fat mass, glucose intolerance, and elevated estradiol and testosterone levels [27,28]. Although the arrest of antral follicle growth and granulosa cell atresia were observed, PCO morphology was not observed in this model. NGF overexpression was to target organs that express 17α-hydroxylase, which is mainly the ovary; however, increased circulating NGF levels may affect systematic sympathetic stimulation. Willson et al. (2014) suggested that the direct effect of NGF or hyperinsulinemia may be the reason for hyperandrogenism in this model. However, given that excess sympathetic activity is considered a reason for obesity as well as insulin resistance [63], a clear relationship among the manifestation of PCOS-like symptoms in this model seems to be elusive.

### 2.3. Diet- or Environmentally or Chemically Induced Animal PCOS Models

To induce PCOS-like phenotypes in laboratory animals, not only genetic but also various other methods have been used (Table 2). We classified the methods into three groups: changing diet (excessive calorie intake), environmental stress, and administration of chemicals.

#### 2.3.1. Diet-Mediated PCOS-Like Phenotypes in Animals

On the basis of the relationship between endocrinal alteration and metabolic controls, animal PCOS models using diet are consistent with obese models. A long-term (14 weeks) high-fat and high-sugar diet (HDHS; high fat chow with 60% of the calories derived from fat and 32% sucrose solution as daily water) significantly elevated the serum testosterone and insulin levels in rats [73]. In addition, irregular estrous cycle, increased fat mass, and cystic ovaries were observed. However, the serum LH level was significantly decreased in this model, which is mostly consistent with previous results obtained using genetic obese animal PCOS models. In addition, the serum estradiol and AMH levels were not different when compared with the control group (normal diet). Therefore, the impaired LH surge may result in the development of cystic ovaries in this model. In a similar study from the same group, rats that received the HDHS diet for 11 weeks manifested hyperinsulinemia but not hyperandrogenemia [74]. In addition, the authors analysed changes in gene expressions caused by the HDHS diet in the ovary; the expressions of ovarian genes related to the primary follicle stage (e.g., epiregulin, *Ereg*), estrogen metabolism (e.g., *Cbr1* or *Ste2*), and insulin receptors (*Insrr*) were mostly shifted. Lai et al. (2014) used a 60% high-fat diet in mice and showed that 20 days of the high-fat diet induced increased serum testosterone levels and an irregular estrous cycle, but no follicular cysts were observed [75]. In this study, the expression ratio of p-Akt/Akt did not change in both liver and skeletal muscle tissue when compared with the control group; however, increased fatty acid uptake and elevated oxidation were observed. In summary, excess calorie intake causes hyperandrogenism, metabolic dysfunctions and PCO features in the end. Bishop et al. (2018) reported that phenotypic alterations in ovarian and uterine structure/function and PCO features were induced by exogenous testosterone in young adult rhesus monkeys fed with a western-style diet [76]. It is possible that prolonged upregulated testosterone levels due to high fat (or high fatty acid) cause both PCO and gonadotropin disruptions.

In another study, a soy-based diet was used; rats reared on the soy-based diet for 28 weeks presented PCO phenotype, prolonged estrous cycle, and glucose intolerance but did not show elevated testosterone levels [77]. Soy is a phytoestrogen, and its estrogenic effects may cause PCO morphology. However, this type of PCO did not cause hyperandrogenism. Serum LH and insulin levels were not measured in this study.

#### 2.3.2. Environmental Changes Induce PCOS-Like Phenotypes in Animals

Since the 1960s, studies on the effects of continuous light exposure on the ovaries of animals have been reported in line with anatomical or surgical interventions to the HPG axis. These studies have been reviewed by Singh [78,79]. Rats exposed to continuous light for several weeks showed PCO morphology and prolonged estrous cycle; the serum LH level was not changed in this method. In addition, a previous study showed that the serum testosterone level was elevated after 16 weeks of continuous light exposure in rats, whereas the body weight was significantly decreased [65]. Given that continuous light abolishes melatonin rhythm, inhibition of the effects of melatonin may induce the symptoms [80]. For instance, prolonged estrous cycle and cystic ovaries were observed in rats that underwent pinealectomy [66]. In hamsters, melatonin is known to depress *Kiss1* expression in the hypothalamus [81]. Therefore, alteration to the HPG axis due to melatonin deprivation may be a reason for developing PCOS-like phenotypes in hamsters. Furthermore, stress-mediated sympathetic nervous activity during constant light exposure could be the cause of PCOS-like features in this model [82]. Although metabolic markers were not analysed in this PCOS model, constant light is known to cause insulin intolerance in rodents and disrupt serum corticosterone levels [83,84]. In summary, several important factors imply that the constant light exposure method induces PCOS in rodents. However, this model has multifactorial effects on animals; thus, it may be difficult to postulate specific causes.

With respect to stressful conditions and sympathetic activity, Bernuci et al. (2008) reported that 8 weeks of chronic cold stress (at 4 °C for 3 h/day) induced PCO morphology and prolonged estrous in rats [67]. Both serum estradiol and testosterone levels were significantly increased. The LH and FSH levels were not statistically different when compared with the control group. In addition, 3 weeks of cold stress upregulated the expression of heat-shock protein 90 (Hsp 90) in the ovary and serum corticosterone levels [85]. Moreover, intraovarian NGF and NE levels were significantly upregulated by cold stress [86]. In combination with the aforementioned 17NF mice, the direct effect of increased sympathetic nervous activity on the ovary is strongly related to the development of PCOS-like phenotypes in rodents. However, non-shivering (adaptive) thermogenesis will occur under cold conditions, which would be also presented in case of excessive calorie intake models, i.e., high-fat diet. Non-shivering thermogenesis is regulated by the thyroid hormone and NE under cold conditions. In this process, fatty acid oxidation occurs actively [87]. Thus, chronic metabolic burdens should not be neglected.

#### 2.3.3. Chemically Induced PCOS-Like Phenotypes in Animals

Chemicals have been used to establish animal PCOS models. Daily subcutaneous injections of d-galactose for 6–7 weeks in mice induced irregular estrous cycle, hyperandrogenemia, and increased serum AMH levels [64]. The PCO morphology was manifested in about 30% of the treated population. Chronic d-galactose administration causes decreased antioxidant activity and increased advanced glycation end products (AGE), which means excessive formation of reactive oxygen species (ROS) in vivo [88,89]. Thus, increased oxidative stress may have a role in altering the functions of the HPG axis.

Neonatal rodents injected with monosodium l-glutamate (MSG) presented unique characteristics in the later adult period. Formation of PCO morphology, irregular estrous cycle, and hyperinsulinemia were observed in the treated mice without elevation of serum testosterone, estradiol, or LH levels [71]. Periovarian fat accumulation and adipocyte size were significantly increased. The degree of AMH staining in the granulosa cells was higher in the MSG-treated mice than in the control group. However, this model may not be similar to PCOS in humans because of normal testosterone levels and severely decreased weight of the ovary and uterus. MSG damages several brain regions, including the arcuate nuclei of the hypothalamus; this results in a significant decrease in the release of the growth hormone and obesity [90,91]. Consequently, alteration of the HPG axis by MSG administration cannot be excluded, e.g., necrosis of *Kiss1* neurons in the hypothalamus. However, the MSG-treated model did not show significantly changed levels of the sex-steroid and gonadotropin hormones, which indicates that the regulatory functions of the hypothalamus seem to remain intact. This model suggests that cystic ovaries with antretic follicles cannot generate hyperandrogenism without LH stimulation. In addition, based on the results, development of PCO morphology may be implicated in metabolic dysfunctions.

Neonatal rats exposed to the well-known endocrine disruptor chemical bisphenol A (BPA) showed PCOS-like phenotypes. Fernandez et al. (2010) reported that consecutive 10-day injections of BPA from postnatal day 1 resulted in the development of the PCO morphology after 4 to 5 months. Serum testosterone and estradiol levels were elevated, with irregular GnRH pulsatility [68]. Recently, another report by the same group indicated that neonatal exposure to BPA alters serum levels of the thyroid-stimulating hormone [92]. In addition, perinatal exposure of rats to BPA results in irregular estrous cycle and reduced serum LH levels [93].

Another endocrine disruptor chemical, tributyltin chloride (TBT), induced irregular estrous cycle and increased testosterone levels by consecutive oral administration to rats. Although the TBT-treated rats did not show PCO morphology, increased fibrosis and apoptosis were observed in the ovary [94]. The TBT-treated rats also presented accompanying metabolic dysfunctions such as increased body weight and elevated serum insulin and leptin levels. The serum estradiol and LH levels were decreased in this model. The authors suggested that the effect of TBT to *Kiss1* neuron actions associated with hyperleptinemia may alter the HPG axis. Leptin deficiency is generally accompanied by downregulation of *Kiss1* expression in the hypothalamus [95,96]; intriguingly, the TBT-treated rats showed downregulated Kiss action with increased leptin concentration.

Letrozole, a P450 aromatase inhibitor, has been widely used for inducing PCOS-like phenotypes in rodents. Kafali et al. (2003) first reported that the administration of letrozole results in the formation of ovarian cysts, acyclic estrous cycle, and elevated serum testosterone and LH levels [69]; the serum estradiol level was significantly decreased. Moreover, prolonged letrozole treatment induced metabolic dysfunctions such as hyperinsulinemia, and insulin intolerance [70]. Moreover, genetic expression was significantly changed in the pituitary gland (increased *Lhb* and *Gnrhr*) and hypothalamus (increased *Kissr1*) of letrozole-treated mice. Macrophage infiltration and elevated mRNA levels of *IL-6*, *TNF-α*, and *MCP-1* were observed in the adipose tissue of letrozole-treated mice [97]. Although some hormonal profiles (e.g., serum estradiol or FSH levels) show variations in different papers, letrozole manifests good reproducibility for PCOS-like features in rodents. This model provides concurrent hyperandrogenism and hypoestrogenism by aromatase inhibition, which is different from other androgen-treated animal PCOS models. However, recovery from the symptoms is observed after the cessation of intervention. In addition, given that no mutations have been reported in P450 aromatase in human PCOS patients and, interestingly, letrozole has been tried for treating anovulation in PCOS patients, searching for the pathogenesis of PCOS by using aromatase inhibitors seems to be somewhat difficult [98,99,100].

Important studies on spontaneously occurring PCOS-like features in non-human primates have been performed. We have added these reports in this section because the pathogenic causes have not yet been determined and environmental factors may not be excluded. Arifin et al. (2008) reported that a cynomolgus monkey (*Macaca fascicularis*) spontaneously developed PCOS-like features [72]. The animal showed a prolonged menstrual cycle, hyperinsulinemia, hyperleptinemia, and elevated androstenedione levels. Moreover, PCO morphology and increased testosterone levels were detected post-mortem; the serum estradiol level was decreased. Abbott et al. (2017) also reported naturally developed PCOS-like traits in rhesus monkeys (*Macaca mulatta*). When they grouped a pool of reared non-human primates by circulating testosterone to high or normal levels, hyperandrogenic monkeys showed statistically elevated serum LH, AMH, estradiol, and cortisol levels when compared with normal testosterone monkeys [3]. The serum testosterone level was correlated with estradiol, LH, and FSH levels, but not serum insulin level, in the naturally hyperandrogenic rhesus monkeys. The endometrial thickness, which was significantly increased in the hyperandrogenic primates, was correlated with the insulin level. In this study, irregular menstrual cycle was not detected, and PCO morphology was not successfully measured in the targeted group. It is expected that further studies with these naturally occurring hyperandrogenic primates will provide us with a better understanding of the pathogenesis of PCOS with respect to genetic or epigenetic aspects.

Because animals are used for studying the pathogenesis of PCOS, naturally occurring polycystic ovaries in animals may provide some insights for the translation of animal data (Table 3). Naturally developed cystic ovaries and hormonal changes in other animal species are not reminiscent of human PCOS; however, some shared mechanisms may exist.

In dairy cattle, the cystic ovarian disease (COD) is a very common reproductive disease, and its prevalence has been reported to be 5% to 30%. COD is a major reason for infertility in cows, and the ovarian cysts are similar to atretic follicles. Vanholder et al. (2006) have reviewed its pathogenesis in cows [101]. Briefly, COD in cows is considered to be induced by the absence or a reduced LH surge at the pre-ovulatory stage. Although the pathophysiology of COD is still unclear, given that most of the cysts are detected at the post-partum stage when energy disturbance is severe, the reduction in the level of insulin or IGFs is strongly related to the development of ovarian cysts. The period of negative energy balance at the post-partum period results in low insulin and IGF levels [102,103]. Reducing these hormone levels is not sufficient to produce enough estradiol in the bovine ovary, which leads to the failure of the negative feedback of estradiol to the hypothalamus. Consequently, gonadotropin release is altered. Cystic cows at this stage show more circulating fatty acids than normal cows. Moreover, metabolic symptoms such as hepatic lipidosis and ketosis were observed in the cystic cows [102]. Recently, altered gene and protein expressions of insulin related factors such as PI3K and IRS-1 were suggested to be attributable for the cystic follicles in the cattle [107]. Thus, metabolic hormones such as insulin and IGF-1 are important regulators of ovarian follicle development as well as cystic formations in dairy cows [108]. Similarly, spontaneous COD was observed in water buffalos, which showed elevated ROS and, conversely, decreased anti-oxidant capacity in the serum and follicular fluid [106]. Therefore, ROS damage to the granulosa cells and imbalance of oxidants/anti-oxidants may be related to the pathogenesis of COD in this animal.

Numerous spontaneously developed polycystic ovaries have been found in pigs. Generally, these ovaries are not accompanied by apparent symptoms, except for persistent estrous and infertility. Although hormone outcomes such as progesterone levels differ depending on the types of cyst follicles and number of developed cysts (polycystic or oligocystic), PCO formation in sows seems to be due to ACTH stimulation and lack of LH release [109]. Thus, COD in pigs is partly related to stress conditions. With respect to androgens, the serum testosterone level was higher in polycystic sows than in normal sows. In addition, the presence of ovarian cysts was correlated to estradiol and testosterone levels in the cystic ovaries [105].

Spontaneous ovarian cysts have also been observed in dogs. Hormonally active ovarian cysts in bitches are the source of hyperestrogenism that presented as alopecia. In addition to the cystic follicles, persistent estrous cycle and cystic endometrial hyperplasia are often manifested in cystic dogs [110]. The estradiol and progesterone levels are correlated to the serum and cystic fluid [104]. Although studies on the pathogenesis of ovarian cysts in dogs are limited, because many of the bitches undergo ovarian hysterectomy in the early ages, it has been proposed that a not high enough LH peak or reduced responsive receptors in the ovary may be related to the pathological development of ovarian cysts.

### 2.4. Evidence from and Implications of the Animal PCOS Models

Although the animal PCOS models and the data from animal studies have several limitations (e.g., hormone and metabolic profiles not evaluated rigorously, no unified standard measures to define the quality and quantity of excess number of dominant follicles, and various time points of the experiments and observation of the samples), the experimental outcomes and suggested or demonstrated biological mechanisms are invaluable for understanding the pathogenesis of PCOS-like manifestations in the animals.

#### 2.4.1. Physiological Insights from the Animal Models with PCOS-Like Symptom

Ovary specific estrogen receptor knockout mice can have elevated testosterone but not pituitary LH levels since negative feedback loops controlling this system are regulated by estradiol and progesterone (Figure 1).

With respect to the regulation of LH release in the animal PCOS models reviewed here, it is conceivable that estrogen or leptin, which is released by white adipose tissue, regulates the activity of *Kiss1* neuron in the hypothalamus [111,112]. The *Kiss1* neuron activates GnRH neurons, and, consequently, LH release from the pituitary gland is increased. In addition, the abrogated inhibitory effect of melatonin on LH release or its frequency may be a reason for hyperandrogenism manifestation. Although serum LH level was not significantly altered in the constant light exposure model and the inhibitory action of melatonin on LH release was recognized only at the neonatal stage, it is possible that melatonin deprivation could enhance LH stimulation by activating *Kiss1* neurons [81,113].

Analysis of the expression of AMH, a member of the transforming growth factor-β family, in the d-galactose and MSG-treated PCOS models has been performed. In these models, serum AMH level or expression in the ovarian tissue was statistically upregulated when compared with the non-treated group. AMH is expressed in the granulosa cells of females, and its expression is correlated to follicular growth and number [114]. In PCOS females, significantly elevated serum AMH levels are observed. Moreover, PCO patients with hyperandrogenism present higher AMH levels than PCO patients without hyperandrogenism [115]. The upregulation of AMH levels in females is suggested as the reason for HPG axis alterations and anovulation [116]. However, further studies on the role of AMH in PCOS are required. In an in vitro experiment, ERβ activation by estradiol suppressed AMH expression, whereas ERα activation stimulated the expression [117]. Thus, discrepancies in the results of the ER KO models have been detected, which indicate that a loss of function of ERα induces the development of PCOS-like features and a loss of function of ERβ is related to the attenuation of PCOS parameters in rodents.

In the case of the animals that showed enhanced sympathetic activity or received cold stress, both types of models presented hyperandrogenism despite decreased or unchanged LH release. Intriguingly, mice overexpressing NGF did not develop ovarian cysts, whereas cystic ovaries were commonly observed in cold-stressed rats. Thus, hyperandrogenism in these models may be mediated by direct nervous input (or stress-mediated inputs) to the ovary, e.g., β_2_-adrenergic receptor activation [118]. In addition, it is unclear whether elevated endogenous androgen synthesis is solely responsible for the development of polycystic ovaries in these studies.

With respect to spontaneously developed PCO in other animals, pigs that present more than 20 ovarian cysts and elevated circulating testosterone levels can be used as a good animal model for PCOS. Although a lack of or immature LH surge is considered the reason for the development of cystic ovaries in animals, including pigs and cows, formation in pigs is partly related to stress-induced (e.g., social or heat stress) dysfunction of the ovary. Increased sympathetic innervation and catecholamines were observed in sows with dexamethasone-induced cystic ovaries [119]. A study by Abbott et al. (2017) showed that female rhesus monkeys with high testosterone levels (about 16% of the total observed monkeys). Thus, animals vulnerable to stress may be prone to developing a PCOS-like phenotypes and infertility.

#### 2.4.2. Molecular Views of the Pathogenesis of PCOS in the Animal PCOS Models

We found some common pathways related to PI3K-Akt signalling (Figure 2). PI3K-Akt signalling is a crucial intracellular signalling for cellular metabolism, proliferation, and survival and glucose homeostasis. Thus, this pathway is emphasized in major disorders, such as cancer, muscular atrophy, and diabetes [120,121,122]. In addition, ovarian follicular development and recruitment are related to this conventional pathway [123]. Recently, the importance of this signalling in the pathogenesis of PCOS was reviewed [124]. Briefly, the PI3K-Akt pathway is activated by insulin, IGF-1, and other growth factors. When insulin or IGF-1 binds to its receptor, insulin receptor substrate-1 (IRS-1) is phosphorylated. Phosphorylated IRS-1 recruits PI3K, which activates Akt, and the activated Akt regulates the activation of several downstream proteins, including FOXO proteins, by phosphorylation. The phosphatase and tensin homolog (PTEN) antagonizes Akt activation and phosphorylation of FOXO proteins. Among the FOXO proteins, upregulated FOXO1 expression, which is mainly expressed in the granulosa cells of the ovary, in cystic follicles and oxidative stress-induced apoptosis in the granulosa cells or rodents are observed [125,126].

It is reasonable to postulate that over-activation of the PI3K-Akt pathway is the most potent reason for the development of PCOS-like phenotypes in the IGF-1 overexpressing mice and tPtenMT mice. A previous study reported that alteration of PI3K-Akt signalling in the ovary, especially the theca cell, occurred in high-fat diet mice [127]; animal PCOS models using the high-fat diet method may have increased PI3K-Akt signalling in the ovary. In addition, β-adrenergic stimulation activates the PI3K-Akt pathway [128,129]. It is unclear whether over-/consistent activation of PI3K-Akt signalling is directly related to hyperandrogenism. Elevated insulin levels are indeed associated with ovarian hyperandrogenism in women with female patients with either type 1 or type 2 diabetes. Their hyperandrogenism is usually mild in comparison to PCOS [38]. Escobar-Morreale has studied and reviewed this in women and points out that it is important not to confuse such causes of female hyperandrogenism with PCOS. and the PI3K-Akt pathway enhances 17α-hydroxylase activity in the ovarian theca cells [130]. Therefore, hyperandrogenism seems to be closely related to PI3K-Akt pathway stimulation. It is noteworthy that metformin, a well-known anti-diabetes drug that has an inhibitory effect on the PI3K-Akt pathway, reduces androgen production in the theca cells and has therapeutic effects on PCOS patients [131,132,133,134,135].

Because phosphorylated FOXO proteins are degraded [136], increased expression of FOXO1 in the cystic ovaries of Zucker rats may partly explain why this model did not show hyperandrogenism, but showed cystic ovaries. Considering the study that described upregulated FOXO1 expression in the cystic follicles of persistent estrous rats, dysregulation of PI3K-Akt signalling may lead to apoptosis of the granulosa cells [40,126]. In addition, the cows probably did not present hyperandrogenism because the development of COD is related to low IGF-1 and insulin levels.

Some of reviewed studies using genetically modified or diet- and chemically induced PCOS models noted that increased free fatty acid (FFA) uptake/lipotoxicity and oxidative stress may have occurred in the ovary tissue (Figure 2). Increased FFA levels and fat accumulation result in oxidative stress in cells and induce impaired insulin signalling [137,138]. High doses of MSG as well as galactose cause cellular oxidative damage [139,140]. Oxidative stress due to ROS also has a potent role in inducing insulin resistance [141]. These events are well-described by many diabetes-related studies [138,142,143]. Briefly, insulin-induced IRS1 tyrosine phosphorylation is inhibited by PKC isoenzyme-mediated serine phosphorylation at IRS1 triggered by FFAs or oxidative stress. Not only FFAs and ROS but also tumour necrosis factor-α and insulin itself can phosphorylate the serine residues at IRS-1. During the serine phosphorylation, insulin signalling is suppressed and related downstream pathways, such as the PI3K-Akt pathway, may be blocked. Therefore, insulin resistance in PCOS may be explained by elevated circulating FFA levels, increased oxidative stress, and chronic stimulation of insulin. However, insulin resistance due to the abrogation of PI3K-Akt signalling would be contrary to the aforementioned lines; hyperandrogenism may be implicated in the activated PI3K-Akt pathway. Nevertheless, it could be accepted that the disruption of the insulin signalling event would have primarily occurred systematically, especially in the skeletal muscles and adipose tissue. Instead, as mentioned above, we postulate that hyperinsulinemia and metabolic dysfunction manifested in the animal PCOS models may be related to ovarian cyst development. We assumed that elevated stimulation of FFAs, oxidative stress, and insulin activate the extracellular signal-regulated kinase (ERK) pathway in the ovary. FFAs, ROS, and insulin promote ERK phosphorylation and downstream cascades [144,145]. Since ERK activation is implicated in cyst formation in vivo, upregulation of the ERK pathway in the ovary may be partly responsible for follicular cyst development mediated by 3′,5′-cyclic adenosine monophosphate (cAMP)-protein kinase A (PKA) signalling [146]. Furthermore, β-adrenergic activation results in the activation of the ERK pathway via cAMP-PKA signalling [147]. Thus, the PCO morphology in the PCOS models with upregulated sympathetic nervous activity could be explained. In addition, upregulated ERK expression in the cystic ovaries of Akt2 KO mice is consistent with this point of view.

### 2.5. Further Directions for Developing and Using Novel Animal PCOS Models

As describe above, molecular pathways related to PI3K-Akt and ERK signalling seem to be highly related to the development of PCOS-like phenotype in animals. In addition, the components of these two pathways interact and are closely connected. The Ras-ERK and PI3K-Akt-mammalian target of the rapamycin (mTOR) pathway regulate each other by cross-inhibition or cross-activation [148]. Moreover, two major signalling pathway both regulates the phosphorylation of FOXO proteins. Considering these signalling pathways are essential for regulating cellular proliferation and metabolism, further studies using chemically or genetically engineered animal models are required to define specific molecular cascades related to the pathogenesis of PCOS. For instance, Hsp 90 expression was upregulated in the ovary of the cold-stressed PCOS model [85]. Hsp 90 is a chaperone molecule that regulates the activity and stability of its client signalling proteins (mainly oncogene-related molecules have been studied). Hsp 90 activates the Akt pathway and indirectly stimulates ERK signalling [149,150,151]. Thus, the role of Hsp 90 in the ovary with respect to PCOS needs to be elucidated.

In addition, it is surprising that few studies focused on PKCs, which are key mediators in overall metabolic dysfunctions, have been performed. PKC activation leads to steroidogenesis in ovarian cells and promotes LH release [152,153,154]. In addition, among the isoenzymes of PKC, loss of function of PKC-θ prevented fat-induced insulin resistance in vivo [155]. Therefore, we expect that alteration of PKC activity and PKC-mediated cellular pathways may result in both hormonal disturbance and metabolic dysfunction in PCOS models.

Non-alcoholic fatty liver disease (NAFLD), in which lipid accumulates in the liver without a history of excessive alcohol consumption, is the most common cause of chronic liver disease. Nowadays, there is an extensive focus on its relationship with PCOS. The prevalence of NAFLD is estimated to be about 10% to 30% in humans [156]. This disorder often manifests as obesity, hyperinsulinemia, and insulin resistance. Thus, because of its similarity and relevance to the metabolic manifestation in PCOS patients, a relationship between the two diseases has been considered and PCOS and NAFLD may have co-existing symptoms [157]. In fact, about 40% of the PCOS patients have NAFLD. Patients with PCOS showed a high prevalence of NAFLD and insulin resistance [158]. The methods for inducing NAFLD in animal models are mostly the same as those discussed in this review, e.g., high-fat diet and genetically modified obese or PTEN KO mice [159]. Therefore, ovarian phenotypes in NAFLD animal models are assumed to be not greatly different to existing PCOS models. However, methionine choline-deficient (MCD) diet, which is frequently used for developing hepatic steatosis in rodents, has not yet been used for studying PCOS in animal models; thus, novel findings with MCD diet-induced NAFLD and PCOS are expected [160].

Recently, studies on gut microbiota that can affect systemic metabolism have been performed because metabolic alteration is strongly related to the development of PCOS phenotypes. Several papers have emphasized that alteration of microbiota was observed in PCOS patients [161,162]. These studies suggested that antigen production by altered gut microbiota may be implicated in metabolic dysfunction in PCOS patients. In addition, changes in gut microbiota are implicated in obesity as well as NAFLD [163]. Given that obesity and metabolic symptoms are important factors related to PCOS, it is important to study the role of microbiota by using animal PCOS models. Early androgen exposure changed microbiota composition in a rat model [164]. Further studies related to this concept need to be performed to evaluate the novel aspects of PCOS pathogenesis.

Non-human primates and pigs, as naturally developed PCOS models, will be useful for tracing spontaneous hormonal changes and determining novel aspects of the manifestation of hyperandrogenism. However, aside from the size, care, and handling problems of the animals, difficulty in detecting the ovarian morphology as well as screening based on serum androgen may limit the use of the animals.

## 3. Conclusions

In this review, we summarized animal PCOS models that used non-direct hormonal interventions to investigate common pathophysiology and molecular mechanism in vivo. Although the manifestation of PCOS symptoms was not fully induced in the reviewed animal PCOS models, we could find some shared pathophysiology and molecular pathways with respect to the PCOS-like manifestations. Hyperandrogenism and polycystic ovarian morphology in the animal models seem to be attributable to different factors, but they are closely connected. On the basis of this perspective, dysregulation of PI3K-Akt and ERK pathways was discussed, and they are cross-regulated. No single factor or disruption of a single molecular pathway causes the hallmarks of PCOS in the animals, resulting in the difficulty in translation from animal data to the pathophysiology of PCOS in humans. Nevertheless, to understand the pathogenesis of PCOS on the basis of scientific manipulation and develop novel treatments, investigation of physiological mechanisms by using animal PCOS models should be continued.

Herein, we could not pay much attention to the HPG axis. It is connected to the central mechanism for PCOS development and is essential for understanding PCOS as an endocrine disorder. However, nearly all perturbations to the animal models, which presented in this review, would eventually affect the HPG axis. Therefore, a lack of careful explanations about the HPG axis, mainly the alteration of neuronal activations, is a limitation of this review.

We expect that this review will be helpful during the selection of animal PCOS models to determine pathophysiological mechanisms and for the development of novel models. We tried to include up-to-date papers and hope that our work would contribute to establish novel animal models. There have been numerous clinical and translational investigational issues that have not been resolved to date [165,166,167,168,169,170,171,172,173]. The efforts to determine the specific mechanisms underlying PCOS will contribute to the development of novel treatments or control methods for this complex syndrome.

## Figures and Tables

**Figure 1 ijms-20-02720-f001:**
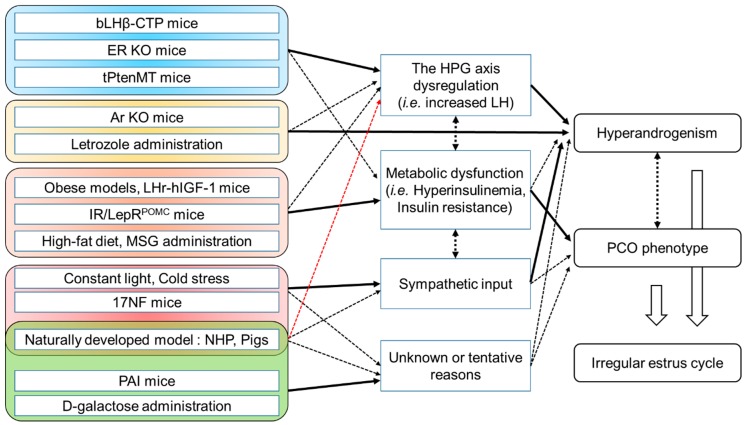
A schematic representation of the summarized pathophysiology to explain the manifestation of elevated testosterone levels and polycystic ovary (PCO) morphology. Solid lines indicate direct effects, and dotted lines, relatively weak or indirect effects. Two-headed arrows present cross-actions.

**Figure 2 ijms-20-02720-f002:**
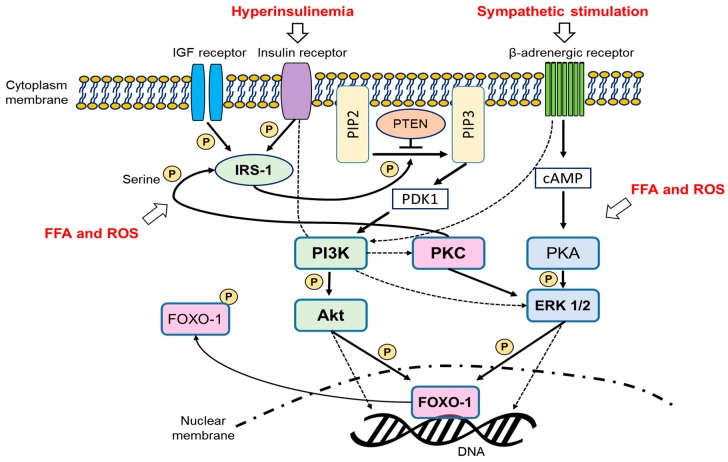
Suggested and integrated molecular signalling possibly related to PCOS development. Solid lines indicate direct effects, and dotted lines, indirect effects. PIP2 = phosphatidylinositol-4,5-bisphosphate, PIP3 = phosphatidylinositol-3,4,5-trisphosphate.

**Table 1 ijms-20-02720-t001:** Transgenic and genetically modified animal models including PCOS-like symptom.

Types	Author	Year	Species	Strain	Estrous Cycle	Ovarian Cyst	Androgen Level	Metabolic Features	LH Level	Time point of Examination (Post-Natal)	Notes
Transgenic	[27]	2009	Mouse	17NF	Prolonged	Yes (+hCG)	↑ (+PMSG)	N/R	↓	28 days	Enhanced sympathetic input; NGF affects trkA receptor and p75^NTR^; apoptosis signaling in the ovarian follicles
Transgenic	[28]	2014	Mouse	17NF	Prolonged	N/R	↑	Obesity, glucose intolerance, hyperinsulinemia	N/A	10, 20 weeks	Increased ovarian sympathetic input
Transgenic	[29]	2010	Mouse	Ar KO	Acyclic	Yes (hemorrhagic)	↑	N/R	↑	19 weeks	Absence of estrogen and elevated LH concentration
Transgenic	[30,31]	1995, 1997	Mouse	LH β-CTP	Prolonged	Yes	↑	N/R	↑	14 - 42 days	The HPG axis dysregulation; Inadequate negative feedback
Transgenic	[32]	2009	Mouse	Theca-specific (CYP17) ERα KO	Irregular	Yes (hemorrhagic, + PMSG or hCG)	↑	N/R	↓	2 - 6 months	ERα KO in theca cells predispose the ovary to develop cysts
Genetic	[33]	2009	Rat	JCR:LA-*cp* strain	Irregular	Yes	↑	Obesity, dyslipidemia, insulin resistance	N/R	6 weeks, 12 weeks	Leptin receptor malfunction; increased insulin and leptin level

# = papers supporting similar study; N.S. = non-significance; N/R = not reported; ↑ = significantly increased/up-regulated; ↓ = significantly decreased/down-regulated; Ar = aromatase; AT2R = angiotensin II type 2 receptor; ER = estrogen receptor; hCG = human chorionic gonadotropin; HPG axis = Hypothalamic-pituitary-gonadal axis; IGF = insulin-like growth factor; LH = luteinizing hormone; NGF = Nerve growth factor; PMSG = pregnant mare serum gonadotropin.

**Table 2 ijms-20-02720-t002:** Diet- or environmentally or chemically induced animal models including PCOS-like symptom.

Types	Author	Year	Species	Methods	Estrous Cycle	Ovarian Cyst	Androgen Level	Metabolic Features	LH Level	Ages before the Intervention	Intervention/Observation Period	Notes
Chemical	[64]	2012	Mouse	D-galactose (S.C.)	Irregular	Yes	↑	N/R	N/R	7–8 weeks	6–7 weeks	Increased AMH level; Formation of ROS and AGEs products
Environmental	[65]	2014	Rat	Constant light	N/R	Yes	↑	N/R	N/R	6 weeks	16 weeks	Altered hypothalamic SCN regulation; Melatonin absence
Environmental	[66]	2004	Rat	Constant light & pinealectomy	Prolonged	Yes	N/R	N/R	N/R	3-4 months	8 month	Melatonin absence; Gonadotropin release dysregulation
Environmental	[67]	2008	Rat	Cold stress circumstance	Irregular	Yes	↑	N/R	N.S.	7–8 weeks	3 h/day, 8 weeks	Increased noradrenergic activity response to cold stress
Chemical	[68]	2010	Rat	Bisphenol A (S.C.)	N/R	Yes	↑	N/R	N/R	-	P1 to 10 treated; 4–5 months (observation period)	GnRH pulse disruption
Chemical	[69]	2003	Rat	Letrozole (P.O.)	Acyclic	Yes	↑	N/R	↑	6 weeks	3 weeks	Elevated testosterone and LH level
Chemical	[70]	2013	Rat	Letrozole (S.C./pellet)	Acyclic	Yes	↑	Insulin resistance, hyperinsulinemia	↑	3 weeks	5 weeks, 10 weeks	High androgen and low estrogen by inhibited aromatase activity
Chemical	[71]	2016	Rat	Monosodium-L-glutamate (S.C.)	Irregular	Yes	N.S.	Obesity, fat accumulation, hyperinsulinemia	N.S.	-	P 2 to 10 treated; P 75 (observation period)	Increased AMH level on the ovarian follicles; hyperinsulinemia
Natural	[72]	2008	Cynomolgus monkey	Naturally occurred	Prolonged	Yes	↑#	Obesity, increased glucose level, hyperinsulinemia, hyperleptinemia	N/R	During >56 months (observation period)		Endometrial hyperplasia with hyperinsulinemia
Natural	[3]	2017	Rhesus monkey	Naturally occurred	N.S.	N/R	↑	N.S.	↑	> 5 years (observation period)		Suggested environmental, epigenetic, prenatally programmed hyperandrogenism suggested

# = papers supporting similar study; * = a review paper; N.S. = non-significance; N/R = not reported; ↑ = significantly increased/up-regulated; ↑# = increased, compared to references; ↓ = significantly decreased/down-regulated; AGEs = advanced glycation end products; AMH = Anti-Müllerian hormone; E = embryonic; GnRH = Gonadotropin-releasing hormone; Hsp = Heat shock protein; P = postnatal; P.O. = oral administration; ROS = reactive oxygen species; S.C. = subcutaneous injection; SCN = suprachiasmatic nucleus.

**Table 3 ijms-20-02720-t003:** Alterations of ovarian phenotype in the other animals.

Types	Author	Year	Species	Methods	Estrous Cycle	Ovarian Cyst	Androgen Level	Metabolic Features	LH Level	Intervention/Observation Period	Notes
Natural	[101]*	2006	Cow	Naturally occurred	Irregular	Yes	N/R	Low insulin and IGF-1 concentration, Ketosis, liver dysfunction	Premature and aberrant LH pulse	-	Hypothalamic-pituitary dysfunction; low insulin and negative energy balance associated metabolic/hormone changes
Natural	[102]	2002	Cow	Naturally occurred	Irregular	Yes	N/R	Free fatty acid ↑	N/R	-	Low IGF-1 leads Hypothalamus-pituitary axis alteration; ketosis and free fatty acid
Natural	[103]	2010	Cow	Naturally occurred	Irregular	Yes	↓	N/R	N/R	-	IGF-II and IGFBP alteration in the ovary tissue
Natural	[104]	2014	Dog	Naturally occurred	Irregular	Yes	N/R	N/R	N/R	-	-
Natural	[105]	2010	Pig	Naturally occurred	N/A	Yes	↑	N/R	N/R	-	-
Natural	[106]	2015	Water buffalo	Naturally occurred	Irregular	Yes	N/R	N/R	N/R	-	Increased reactive oxygen species and decrease antioxidant capacity

* = a review paper; N.S. = non-significance; N/R = not reported; ↑ = significantly increased/up-regulated; ↓ = significantly decreased/down-regulated.

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
