# Peer review of "Animal Models for Human Polycystic Ovary Syndrome (PCOS) Focused on the Use of Indirect Hormonal Perturbations: A Review of the Literature"

_ijms, 2019, doi:10.3390/ijms20112720_

Round 1
Reviewer 1 Report
This paper provides a review of animal models that can help dissect improve understanding of the pathogenic mechanisms underlying PCOS in women. Once the authors reach the latter part of the discussion, the core novel synthesis of their review suggesting potential new understanding of pathogenic mechanisms contributing to components of PCOS becomes clear. Until then, the review feels too much like a re-hash of some recently published reviews on PCOS animal models, but less well focused. To provide a useful contribution to the contemporary literature, the authors need to concentrate their examples and argument in better leading the reader towards their novel models and mechanistic consideration.
l.80, 82. The terms “cystic follicles” or “cystic ovary” are inappropriate for PCOS. Polycystic ovaries grow too many non-dominant antral follicles that undergo atresia before becoming dominant. Animal models with cystic structures do not emulate PCOS. Those with excess numbers of antral follicles, on the other hand, do emulate PCOS.
l.88. Hyperandrogenism is a major clinical hallmark of PCOS, but since there is the normo-androgenic phenotype for PCOS, it is not “the” hallmark.
l.99. Point out that 3-week old female mice are pre-pubertal.
l.109. LH pulse frequency is also elevated in prenatally androgen exposed mice, sheep and monkeys.
l.112. Silva’s 2018 paper addresses a more contemporary understanding of neuroanatomical changes from androgen exposure.
l.113. Both prenatally androgen exposed sheep and monkeys show evidence of ovarian morphology suggestive of polycystic ovaries. This is not true of mouse and rat equivalents.
l.127-130. In fetal sheep and monkeys, androgen exposed females only experience fetal male levels of testosterone. There is no “excessive” exposure of “males and females”. In addition, newborn daughters of women with PCOS demonstrate a reliable biomarker of fetal androgen exposure, elongated anogenital distance. This also occurs in adult women with PCOS. Other biomarkers of fetal androgen exposure are also found in PCOS daughters or women with PCOS. To discount PCOS pregnancies in humans as devoid of increased androgen exposure reflects bias in selection of literature reviewed.
Table 1. To be a model for PCOS, the animal studied has to exhibit at least two of the three diagnostic criteria. Elevated circulating AMH levels could provide a surrogate measure for increased numbers of antral follicles. Such considerations only apply to studies 1, 2, 4, 5, 10 and 17. For example, study 6 animals have regular cycles and are not hyperandrogenic. They have no relevance for PCOS. Obesity per se is not a precursor to PCOS or all obese women would have PCOS. The text at l.158-162 makes this very point. The authors cannot therefore include such animals as “models of PCOS”.
l.168-194. These paragraphs are more informative as they dissect out potential causative and non-causative factors contributing to PCOS-like traits. To call all such models “models for PCOS”, however, is misleading. Contrasting different phenotypes related to different genetic manipulations is a useful exercise, but by definition it means that some of the models are not PCOS-like.
Table 2. Analogous issues to those found with Table 1.
l.356-360. It is unclear what the authors mean here.
l.415-436. Group all naturally occurring PCOS-like animal studies together. They are currently inappropriately split between chemical causes and natural causes. All could be caused by environmental exposures (captive and free-living) or none could be caused by chemical exposures.
Table 3. Cystic ovaries in cattle are not related to PCOS and the underlying mechanisms have been well described by Wiltbank. Morphological presentation is not polycystic ovary-like.
l.483-484. The authors inappropriately group together cystic and polycystic ovaries, and thus arrive at a potentially erroneous conclusion that polycystic ovaries and hyperandrogenism arise separately. The authors need to carefully define ovarian morphology (or AMH physiology) emulating PCOS ovaries and morphology that does not, i.e., cystic or hemorrhagic ovaries. They eventually engage AMH in this manner by l.511-520, but it needs to be connected throughout.
l.489. In humans, in stark contrast to rodents (and possibly other non primates), increased lipid accumulation or BMI inhibits LH release, hence why LH levels in obese women with PCOS can be closer to normal than in lean/overweight women with PCOS. The authors need to be careful of this when trying to dissect out LH-neuroendocrine vs insulin-adiposity mechanisms.
l.496-497. Ovary specific estrogen receptor knockout mice can have elevated testosterone but not pituitary LH levels since negative feedback loops controlling this system are regulated by estradiol and progesterone.
Figure 1. At least naturally occurring PCOS-like monkeys in Table 2 are shown as having high LH levels so there needs to be some arrow connection between those two boxes here to reflect Table 2.
l.530-532. Altered thyroid hormone release is a basis for exclusion from PCOS in women. The authors must make the same distinction clear in dealing with animal model considerations. The mechanistic underpinnings are not the same.
l.531. The authors confuse elevated cortisol levels with stress. Altered adrenal glucocorticoid release can occur for a variety of reasons, including altered steroidogenic enzymatic function and regulation that are not based in stress.
l.567-569. Elevated insulin levels are indeed associated with ovarian hyperandrogenism in women with female patients with either type 1 or type 2 diabetes. Their hyperandrogenism is usually mild in comparison to PCOS. Escobar-Morreale has studied and reviewed this in women and points out not to confuse such causes of female hyperandrogenism with PCOS.
Author Response
Please read the attached file.

Reviewer 2 Report
The authors review the PCOS animal models and indicate their PCOS-like symptoms. They also address the pathophysiology to explain the manifestation of elevated testosterone and PCO morphology. Moreover, they show a molecular signaling possibly related to PCOS development. This review shows the pros and cons of each PCOS models that help researchers to choose suitable models for further investigations of PCOS. I only have minor suggestions that may improve the manuscript:
1. The reference number should be added in Table 1-3.
2. Table 1 and 3 should be rearranged following the species.
3. Please clarify the molecular signaling occurred in which tissue or cell type in Figure 2.
Author Response
Please read the attached file.

Round 2
Reviewer 1 Report
This first revision of a review examining animal models of PCOS, while extensive, is not well written. Arguments can be difficult to follow because of a mixture of faulty reasoning and poorly expressed writing. Points made can be disconnected at times because they arise at different places in the manuscript, and grammar and sentence construction need attention throughout the manuscript. While some of the previous review concerns were addressed, others were not. In sum, this is still a confused review that requires better integration and refinement of arguments.
l.67-68. Despite recognizing naturally occurring PCOS traits in monkeys in Table 2, the authors still misleading state that these do not occur “in rodents and other animals”. They need to cite at least Abbott 2017 and their Table 2 here documenting naturally occurring PCOS traits.
l.72. The uses of a variety of transgenic rodent strains have many limitations. Do not bias selection towards “non-human primates” for such unsubstantiated negative consideration on p.1.
l.131-132. The authors in this paper, and many other authors worldwide in peer-reviewed publications, have repeatedly reported PCOS-like symptoms in female offspring prenatally exposed to androgens. The authors of this paper, therefore, cannot state here that “it is unclear whether these methods induce the PCOS-like symptoms”.
l.138. The AMH mouse model produced a very “distinct phenotype”. The authors’ assertion to the contrary here is incorrect.
l.143-145. The authors’ argument here makes no sense. Their reason for discounting models of “direct hormonal intervention” is because they cannot understand “from where the excessive circulating androgens arise”. Hyperandrogenic PCOS pregnancies in women are well documented. In the last several years, increasing numbers of publications document biomarkers of a hyperandrogenic intra-uterine environment for daughters of women with PCOS. Sixty percent and more of such daughters grow up to have PCOS themselves. A condition can therefore be documented to exist without it the source or sources of pathogenesis being known.
p.8, l.10-11. Therapeutic insulin in type 1 diabetes and hyperinsuinemia in type 2 diabetes do generate a degree of hyperandrogenism in women. Escobar-Morreale has reported on this. The authors need to emphasize this species difference more than they currently do here. Insulin may indeed drive high androgen levels in women even if it fails to do so in female rodent models. The authors bring up this very point later at p.20, l.113-115, but do not connect it with the relevant point here.
p.14, l.21. The authors missed the opportunity here to include the peri-pubertal onset, hyperandrogenism-induced nonhuman primate model for PCOS reported in several publications by Bishop, True and colleagues. They nicely show the additive effects of androgens and high calorie diet on PCOS-like symptomology.
Figure 1. The added line between the box containing “naturally developed model: NHP and pig” and the box containing HPG axis and increased LH shown in the authors’ response letter is still absent from this Figure in the actual manuscript. Showing it as a dashed line would be appropriate.
p.20, l.85-88. The two sentences in red text bear no relationship to each other. The sentence concerning monkeys in the Abbott 2017 study correctly does not mention stress. Therefore to include this latter sentence in a section devoted to stress in PCOS animal models is without basis. The authors should read the 2003 paper by Saplosky, Abbott and others on the inconsistent linkage of relatively high cortisol (but not hypercortisolemia) with stress, especially in nonhuman primates.
Round 3
Reviewer 1 Report
The authors have sufficiently responded to previous review concerns. I have no remaining issues.